# Describing the development and implementation of a novel collaborative multidisciplinary approach to deliver foot health supports for individuals experiencing homelessness and its outcomes

Rajna Ogrin[1]*, Mary-Anne Rushford[2], Joseph Fallon[2], Rebecca Mannix[3], Ben Quinn[3], Anthony Lewis[4]

1 Bolton Clarke Research Institute, Forest Hill, Victoria, Australia, 2 Bolton Clarke Homeless Persons Program, West Melbourne, Victoria, Australia, 3 cohealth, Melbourne, Victoria, Australia, 4 Footscape, Macleod, Victoria, Australia

* rogrin@boltonclarke.com.au

## Abstract

Basic foot care is a real need of people experiencing homelessness. To improve access to foot health for this group, three services structured to provide healthcare support for people experiencing homelessness collaborated in metropolitan Melbourne, Australia: an established nurse-led Homeless Persons Program (HPP), a specialty community health podiatry clinic servicing people experiencing homelessness, and a charity supporting disadvantaged communities providing free socks, foot first aid kits and second-hand footwear for distribution by nurses and podiatrists of participating services. This paper outlines the implementation and evaluation of this collaboration. A four stage implementation approach was used, addressing: 1. *Who needs to do what differently*? 2. *What are the barriers and enablers that need to be addressed*? 3. *Which intervention components could overcome the modifiable barriers and enhance the enablers*? 4. *How can the changes be measured*? The evaluation prospectively collected information about how HPP nurses referred adults to podiatry, and whether the referred individuals accessed the podiatry clinic, the outcomes of the podiatry visit, and how many received footwear, socks and foot first aid kits provided by the non-profit organisation. Over 1$^{st}$ June 2019 and 31$^{st}$ December 2020, 52 individuals were identified as adults who could potentially benefit from podiatry by the HPP nurses, of which 33 accessed podiatry. Those who did not visit the podiatry clinic were more likely to be born outside of Australia, live in more precarious housing (crisis accommodation and rough sleeping), have slightly more predisposing factors for homelessness, but have fewer medical, psychological and cognitive conditions. A structured approach including processes, education, regular, outreach to youth refuges and formal outcome monitoring enabled foot health care access in people experiencing homelessness. Further research is needed to ascertain how to support participants at risk of foot problems to access podiatry before their foot health issue reaches crisis point.

**Data Availability Statement:** Data cannot be shared publicly because of high level of confidentiality requested by participants - they are very concerned about data sharing. Access to data will require approval from the study Ethics Committee. The Ethics Committee can be contacted at: ethics@boltonclarke.com.au. Those interested can also contact lead author Dr. Rajna Ogrin at rogrin@boltonclarke.com.au for further information.

**Funding:** The author(s) received no specific funding for this work.

**Competing interests:** The authors have declared that no competing interests exist.

## Introduction

Homelessness is a significant issue in cities worldwide. Homelessness is not 'roof'lessness (defined as the absence of a roof over one's head). It is more than 'sleeping rough' and includes living in marginal and insecure accommodation such as couch surfing, rooming houses or crisis accommodation [1]. Homelessness also includes people who are socially excluded and have minimal independent resources or capacity to self-support [2]. In Victoria, Australia there are approximately 24, 817 people currently experiencing homelessness [2, 3].

Foot problems are a common health issue, affecting between three and 27% of the general population [4–6] and, for individuals experiencing homelessness, affecting between nine and 72% [7, 8]. Most common foot problems in this group are tinea pedis, foot pain, functional limitations with walking, and improperly-fitting shoes [7]. Unique issues face this population including wearing inappropriate shoes, prolonged contact with moisture, long days of standing/walking in wet shoes and exposure to extreme cold, resulting in poor foot health [8].

General practice is a common first access point to care for people experiencing foot problems [4]. In many western countries, specific footcare provision can be provided by private or publicly funded allied health care providers specifically trained in assessing, diagnosing and managing the feet, such as podiatrists or chiropodists. Individuals experiencing homelessness encounter barriers towards accessing mainstream foot care [7]. These barriers include the limited availability of affordable and accessible foot care services, embarrassment about their socioeconomic status or their foot condition, and their lack of permanent and affordable housing options which may prevent them from seeking care [7, 9]. While research in foot health in people experiencing homelessness is limited, previous research suggests that targeted efforts to screen for and treat foot problems could result in improved health and social outcomes for individuals experiencing homelessness [7]. Specifically, proper fitting shoes would contribute to decreased pain, fewer pain locations, improvements in foot-related health, and faster walking speed in this group [10]. Further, the development of a foot care model in people experiencing homelessness in Canada outlines the need for coordinating care [8]. However, there is a paucity of literature available on how to implement a program to support access to foot care by people experiencing homelessness.

In 2019, a charity called Igniting Change interviewed individuals experiencing homelessness to understand their needs. The aim was to identify gaps in services in the Central Business District (CBD) of Melbourne, Australia. Many of the individuals interviewed reported they experienced problems with foot health and accessing footwear. This led to the charity donating $5000 to support increased access to footwear and podiatry services for individuals experiencing homelessness in the CBD of Melbourne. It also led to the initiation of a collaboration between a community health nurse outreach team, a podiatry service that prioritises people and communities who experience social disadvantage, and a charity providing foot health resources in the 'Happy Feet' program.

Implementation Science is a field of research that considers what is needed to improve uptake of evidence into practice [11]. The implementation of new programs are likely to be more successful when underpinned by theoretical approaches [12], with many implementation frameworks developed to guide new program implementation [13]. Further, the implementation of any new service requires care providers to change their behaviour [14]. A common barrier for uptake of programs to improve health, includes the target health issue not being front of mind of healthcare staff [15]. Further, programs supporting people experiencing homelessness are commonly fragmented [8], where there are no structured processes or coordination in place to enable identification, referral, and monitoring between existing organisations who provide foot health support and have experience in working with structurally disadvantaged

groups. This article focuses on describing the development of the Happy Feet program. It includes the description of the service delivery elements necessary to implement the new program, the individuals who were assessed as needing foot care and their access to podiatry and foot health resources.

## Methods

This study reporting follows the Standards for reporting implementation studies (StaRI) [16].

### Study design and aims

The aim of this study was to describe the development, implementation and utilisation of the novel Happy Feet program in metropolitan Melbourne.

### Setting

This project was undertaken within the City of Melbourne local government area, a large metropolitan, multicultural city centre in Australia. Melbourne comprises a 37.7km$^2$ area, has a residential population of 183,756 with a median age of 28 years and 56% of residents are born overseas. In 2018, 1728 residents of Melbourne were reported as experiencing homelessness [3].

### Participants

Eligible participants for this project included those who were 18 years or older, English-speaking, receiving services from the Homeless Persons Program (HPP, described in Box 1) and were identified as benefiting from foot health support, ie. participants who had skin lesions, nail abnormalities, complained of foot pain or had other foot issues that would benefit from podiatry treatment. Participants who did not meet these criteria were excluded.

Participants were eligible for the HPP program if they were experiencing or at risk of homelessness. HPP uses a broad definition of homelessness, where homelessness may mean: being socially isolated without the support network of family and friends; the absence of a roof over one's head; or living in unstable or unsafe housing [18].

### Happy Feet program implementation

To develop this new program, the Happy Feet Program Implementation Team (The Team) was formed, including members from each collaborator: a. The age and community care organisation: The HPP manager and nurse champion, and a senior researcher from the organisations research institute; b. Community health organisation podiatrist providing foot care and her manager; c. Charity providing foot health resources: the Chief Executive Officer (CEO). The Team adapted a four step implementation approach [19], heavily influenced by the work of Lavis et al's Knowledge Transfer Framework [20], selected due to its clear and practical methods to promote behaviour change by staff. Elements of the Implementation Framework of Aged Care [21] were also utilised, namely the engagement of key stakeholders across the whole implementation process. The following steps were undertaken:

**Step 1. Who needs to do what differently?.** The specialist podiatry service identified that few people were accessing their services, despite there being high rates of homelessness and that foot problems are common in this population group. The HPP staff recognised that many of their clients may benefit from referrals to podiatry, but they undertook few referrals to this program. Therefore, the aim of the Happy Feet Program was for HPP nurses to identify

Box 1. The Happy Feet program components.

### Community Health Nurse Outreach

This was delivered by Bolton Clarke, a not for profit aged and community care provider of independent living services through at-home care, retirement living and residential aged care. Bolton Clarke Homeless Persons Program (HPP) aims to promote the health of individuals and families experiencing homelessness. The program is underpinned by the Social Model of Health and Primary Health Care Principles [17]. In operation since 1978, HPP operates using a rights and equity-based model of health care, which is underpinned by the belief that people who experience homelessness have the right to holistic health care that is accessible and relevant; of a high standard; equivalent to that received by the general community; and self-determined. HPP offers an assertive, primary health care response either a. fixed outreach in the form of a nursing clinic co-located with a community agency or accommodation service or, b. mobile outreach or Community Health Nurses (CHN) outreaching to public places, food programs, rooming houses, caravan parks, supported accommodation and other homeless services.

### Podiatry

This was delivered by cohealth, one of Victoria's largest not for profit community health services with a mission to improve health and wellbeing for all, and to tackle inequality and inequity in partnership with people and their communities. cohealth prioritises people and communities who experience social disadvantage and provides a range of health and other supports to people experiencing homeless. cohealth, in partnership with City of Melbourne, has been providing podiatry services to people who are experiencing or at risk of homeless or at risk of homelessness since 2012.

### Foot health resources

This was provided by Footscape, a charitable organisation that recognises disadvantaged communities are predisposed to debilitating foot pathology. Working in partnership, Footscape assists to provide foot health resources and evidence-based care across Victoria.

individuals experiencing homelessness who have foot health needs and refer them to podiatry care and/or provide foot health resources.

**Step 2. What are the barriers and enablers that need to be addressed?.** The Team met to discuss the barriers and enablers to target behaviours and guide the choice of intervention components. The barriers were identified through talking to staff, where foot health was recognised as not being front of mind of HPP nursing staff. The identified enablers were: a. That the HPP nurses were aware of the podiatrist; b. Both the HPP nurses and podiatrist had developed trusting relationships and were confident to refer/accept the target individuals; and c. All of the staff/collaborators involved were sensitive to the needs of people experiencing homelessness, understanding how to engage with this group to engender trust. Sensitivity to needs and engendering trust are crucial for engagement with this group [22–24]. The Team considered activities to address all of these barriers (lack of awareness and coordination) and enablers

**Table 1. Activities that were developed to enable a foot health collaboration in the Happy Feet program.**

| Activity | Details | Barrier addressed (B)/ Enabler supported (E) |
|---|---|---|
| Selection of a Champion | Selection of a champion from the HPP Team to be the point of contact for all staff. | Improving coordination between services (B) |
| Development of processes for foot health support | Individuals experiencing homelessness were identified to receive foot health resources and/or be referred to podiatry and access podiatry care. The processes are outlined in Fig 1. | Improving coordination between services (B) |
| Education | One hour education of nurses by the podiatrist to identify foot issues | Increasing awareness of podiatry by HPP nurses (B) and consolidating trusting relationship between frontline staff of collaborators (E) |
| Regular meetings | Managers from each organisation to discuss activities and progress. | Supporting coordination between services (B) and consolidating trusting relationship between collaborators (B) |
| Regular updates to frontline staff | Monthly meetings where the program was discussed to keep foot health front of mind for frontline staff; regular emails from the champion, reminding them of the program. | Increasing awareness of podiatry by HPP nurses (B) |
| Outreach activities | Engaging youth at youth refuges, to increase uptake of foot health activities in this population. | Increasing awareness of podiatry by HPP nurses (B) |
| Monitoring of outcomes | Formalised mechanism to assess the impact of the activities on access to supports, feeding back to staff. | Increasing awareness of podiatry by HPP nurses (B) |

(trust between collaborators, and with people experiencing homelessness) within the Happy Feet program.

**Step 3: Which intervention components could overcome the modifiable barriers and enhance the enablers?.** The Team met and developed six activities to support the uptake of foot health to enable the delivery of The Happy Feet program, shown in Table 1. To further increase access, all services were provided at no cost to the individual. A description of the program and each collaborator involved in the foot health support team is shown in Box 1.

**Step 4. How can the changes be measured?.** The Team discussed the best way to ascertain that the changes made a difference, and proposed the following outcome measures to be captured as part of the evaluation:

*Primary outcome measure.* The number of people experiencing homelessness referred by HPP nurses and accessing foot health resources and/or community health podiatry.

*Secondary outcome measures.* The number of attendances to podiatry; treatment needs; treatment outcomes; reach, adoption, fidelity and sustainability of the service implementation.

## Data collection: Participant outcomes

Fig 1 demonstrates the process and time points for intervention data collection during the study. Participants were invited to consent to participate in this project and asked to complete a participant information and consent form. Once consent was obtained, the HPP nurses completed a data sheet that included basic demographic information, information on housing status, physical categorisation using International Classification of Disease (ICD -10) [25] mental and cognitive conditions, and to outline the participant's foot health issues that would benefit from podiatry review. Predisposing factors for homelessness were also collected, as this data is routinely collected by the HPP nurses to identify factors that increase risk for experiencing

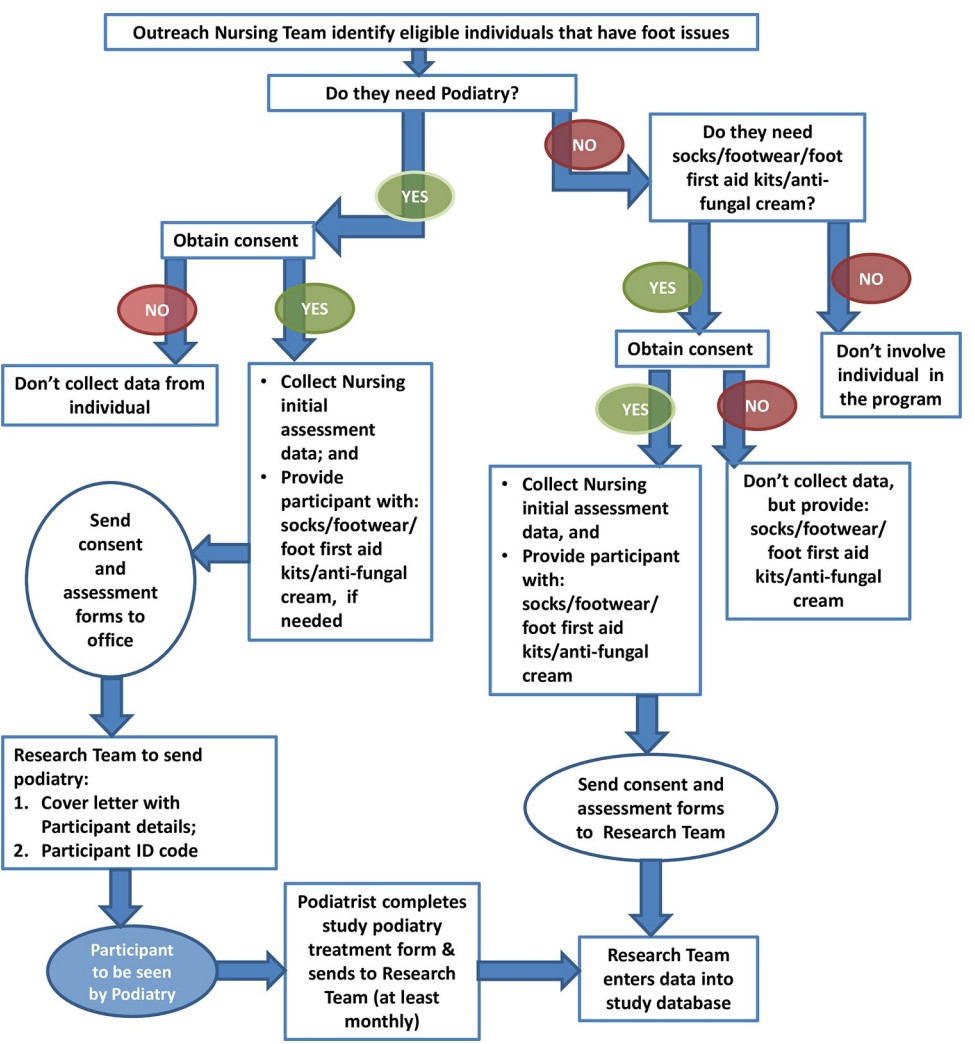

**Fig 1. Participant journey and intervention data collection in the study.**

homelessness. This data includes whether the individual is experiencing mental and/or physical health issues, social issues, alcohol or drug issues, behavioural issues or intellectual/cognitive issues [18]. Data was also collected on whether any foot health material aid was provided, and whether the participants were willing to be referred to podiatry. All data was deidentified, where each participant was allocated a code after consent was obtained. Data was collected on coded paper-based forms.

Participants underwent a foot assessment when seen by the podiatrist, and the podiatrist collected data on a standardised paper-based form. This included the date of the appointment, whether the participant had cancelled or did not attend previous appointments, the reason for referral provided by the individual, and then the foot health issues identified by the assessment. Foot health issues included skin and nail issues, foot function issues, presence and extent of pain, peripheral sensation and blood supply. The podiatrist documented the treatment provided, the impact of the treatment on the participant in their own words, and whether foot health material aid was provided.

## Data collection: Implementation outcomes

Process data on the implementation of the Happy Feet Program was collected, using the existing outcome taxonomy that encompasses constructs of reach, adoption, feasibility, acceptability, appropriateness, fidelity and sustainability [26]. To ascertain the reach and adoption (does it reach the right people and are they using it), demographic data of participants, the number who engaged with the HPP Nurse and podiatrist was collected prospectively by the staff of the respective services. To ascertain the feasibility, acceptability and appropriateness of the program, feedback at the regular HPP staff meetings was collected. To ascertain the fidelity and sustainability, data on the delivery of the program components was collected prospectively by HPP and podiatry staff. Changes in organisational support systems and processes was also collected, documented throughout the project by the researcher and in the minutes from the regular meetings held by the key stakeholders from the three collaborating organisations.

Ethics approval was provided by the Bolton Clarke Human Research Ethics committee (approval number 170011) and Human Research Advisory Group of cohealth (project number 1808), and all participants provided written consent. Data collection was originally across a 12 month period. Recruitment of people experiencing homelessness in research is always incredibly difficult due to issues of trust, and the transient nature of this group [23]. This study was no different, and due to low numbers recruited into the study, the recruitment was extended and occurred between 1st June 2019 and 31st Dec 2020.

## Data interpretation and analysis

Data is presented descriptively. Due to the small numbers, continuous data is presented as medians and interquartile ranges. Categorical variables are presented as frequencies and percentages. Data was analysed using SPSS version 25 (IBM, USA).

## Results: Participant outcomes

### Participant demographic information

52 individuals were included in the study over the 19 month period, with a mean age of 41.48 (±15.46) years; an additional two individuals underwent assessment, but were excluded from the data analysis due to being under age. There were a greater number of male than female participants, and most were Australian born, with countries of birth outside of Australia including Columbia (n = 1), Ethiopia (n = 1), India (n = 1), Korea (n = 2), Somalia (n = 1), South Sudan (n = 4) and Tonga (n = 1). The primary language spoken for the majority of participants was English (n = 45), with other primary spoken languages being Arabic (n = 1), Korean (n = 2), Somali (n = 1), Spanish (n = 1), Turkish (n = 1) and Urdu/Hindi (n = 1). Additional details are shown in Table 2.

Of the 52 participants referred to podiatry care by the HPP nurses, 28 underwent a podiatry assessment, five accessed podiatry education but did not undergo a podiatry assessment, and 19 participants did not access the podiatrist after the initial nurse contact and referral to podiatry. The five participants who accessed podiatry education but did not undergo a podiatry assessment were seen at a youth centre or refuge (classified as outreach podiatry). Information describing all participants and access groups is shown in Table 2. Those participants who did not access podiatry were more likely to live in more precarious housing (rough sleeping), have slightly more predisposing factors for requiring HPP nursing care, but have fewer self-reported medical, psychological and cognitive conditions.

**Table 2. Details of participants recruited into the project.**

| Factor | Accessed podiatry clinic (N = 23) | Accessed outreach podiatry (N = 10) | No podiatry accessed (N = 19) | Total participants (N = 52) |
|---|---|---|---|---|
| **Age years Median (IQR)** | 50.00 (40.0, 54.5) | 21.00 (20.5, 23.25) | 39.00 (34.0, 52.0) | 41.50 (27.75, 52.0) |
| Missing n(%) | 2 (9) | 0 | 0 | 2 (4) |
| **Gender n(%)** | | | | |
| *Female* | 10 (44) | 7 (70) | 5 (26) | 22 (37.5) |
| *Male* | 13 (57) | 3 (30) | 13 (68) | 29 (58.3) |
| *Transgender* | 0 | 0 | 1 (5%) | 1 (2%) |
| **Country of Birth n(%)** | | | | |
| *Australia* | 19 (83) | 3 (30) | 14 (74) | 36 (69) |
| *Other* | 2 (9) | 7 (70) | 3 (16) | 12 (23) |
| **Missing n(%)** | 2 (9) | 0 | 2 (11) | 4 (8) |
| **Primary Language spoken n(%)** | | | | |
| *English* | 20 (87) | 8 (80) | 17 (89) | 45 (87) |
| *Other* | 3 (13) | 2 (20) | 2 (11) | 7 (13) |
| **Accommodation Type n(%)** | | | | |
| *Couch surfing* | 2 (9) | 0 | 1 (5.3) | 3 (6) |
| *Crisis accommodation* | 1 (4) | 2 (20) | 3 (16) | 6 (12) |
| *Hotel* | 1 (4) | 0 | 0 | 1 (2) |
| *Private residence* | 0 | 1 (10) | 0 | 1 (2) |
| *Private rooming house* | 4 (17) | 0 | 1 (5) | 5 (10) |
| *Public housing* | 6 (26) | 1 (10) | 3 (16) | 10 (19) |
| *Public rooming house* | 1 (4) | 0 | 0 | 1 (2) |
| *Refuge* | 0 | 4 (40) | 1 (5) | 5 (5) |
| *Rough sleeping* | 8 (34) | 0 | 9 (47) | 17 (33) |
| *Supported Residential Service* | 0 | 0 | 1 (5) | 1 (2) |
| **Missing n(%)** | 0 | 2 (20) | 0 | 2 (4) |
| **Number of predisposing factors to homelessness (Median (IQR))** | 3.00 (2.0, 3.0) | 1.50 (1.0, 2.25) | 3.00 (2.0, 4.0) | 3.00 (2.00, 3.00) |
| **Total medical issues** Median (IQR) | 1.0 (0.75, 2.0) | 0.00 (0.0, 1.0) | 1.0 (1.0, 1.75) | 1.00 (0.00, 2.00) |
| Missing n(%) | 1(4) | 0 | 3(16) | 4(8) |
| **Total mental health issues** Median (IQR) | 1.00 (1.0, 2.0) | 1.5 (0.00, 2.50) | 1.00 (0.50, 2.50) | 1.00 (1.00, 2.00) |
| Missing n(%) | 0 | 0 | 2(11) | 1(2) |
| **Total cognitive issues** Median (IQR) | 0.00 (0.00, 0.00) | 0.00 (0.00, 0.00) | 0.00 (0.00, 0.00) | 0.00 (0.00, 0.00) |
| Missing n(%) | 0 | 0 | 1(5) | 1(2) |
| **Total number of combined issues (medical, mental and cognitive)** Median (IQR) | 2.00 (2.00, 3.00) | 2.00 (0.75, 3.25) | 2.50 (1.75, 3.00) | 2.00 (2.00, 3.00) |
| Missing n(%) | 0 | 0 | 1(5) | 1(2) |

### Foot specific information: From HPP nurses

The primary reasons for podiatry referral by the HPP nurses are shown in Table 3, with the most common reasons being general nail and callus treatment, general foot assessments, skin issues, pain and presence of diabetes.

**Table 3. HPP nurse reason for podiatry referral (n = 52) and podiatrist documented reason for participant attendance (n = 25).**

| Reasons for podiatry referral | Nurse | Podiatrist |
|---|---|---|
| | Frequency | Frequency |
| | N = 52 | N = 25 |
| | n(%) | n(%) |
| General nail and callus treatment | 14 (27) | 15 (54) |
| General foot assessment | 9 (17) | 3 (11) |
| Skin issues including tinea | 7 (13) | - |
| Joint deformity/Pain in the foot/ankle | 7 (14) | 4 (15) |
| Type 2 diabetes and foot health | 5 (10) | - |
| Foot wounds and serious foot complications (+/- diabetes) | 5 (10) | 2 (7) |
| Peripheral neuropathy | 1 (2) | - |
| Self-harming of feet | 1 (2) | - |
| Shoe advice/Requires footwear | 1 (2) | 1 (4) |
| Swollen legs—requested podiatry | 1 (2) | - |
| Missing | 1 (2) | 3 (11) |
| **Total** | **52** | **28** |

Information on HPP nurse provision of the free foot health resources is shown in Table 4. The HPP nurses referred seven participants to the following other services: Emergency Department to remove glass from their foot; to a GP for referral for bunion surgery; physiotherapy referral (n = 2); and local (mainstream) podiatry.

## Podiatry specific information

28 of 52 (54%) participants accessed the podiatrist and had a foot assessment. The median (IQR) days between nurse referral and accessing the podiatrist was 3.5 (0–17.25) days. 23 of the 28 podiatrist interactions occurred at the clinic, with five contacts occurring as outreach. Of the 23 who attended the podiatry clinic, four participants missed their scheduled appointment and rescheduled. Only one participant had more than one podiatry appointment at the clinic during the study period (n = 2 appointments).

The podiatrist documented the reason for the visit for 25 participants, shown in Table 3. The most common reason was skin and/or nail issues (54%). 15 (54%) were documented as having the foot issue long-term, and three (11%) short term, although two participants were documented as not relevant, with data missing for eight (29%) participants.

## Foot assessment findings

Over half of the participants assessed by the podiatrist had nail pathology, abnormal pressure, corns and/or callus, deformity or bony prominences. Severe foot issues were identified, with

**Table 4. Provision of foot health resources by nurses and podiatrist.**

| Foot health resources provision | Provision by Nurses | Provision by podiatrist |
|---|---|---|
| | n (%) | n (%) |
| Shoes | 8 (15) | 14 (50) |
| Socks | 10 (19) | 17 (61) |
| Foot First Aid Kid | 6 (12) | 11 (39) |
| Anti-fungal cream | 0 | 3(11) |

three participants having an acute wound, one participant having a chronic wound, one a current foot infection, and pain was present in seven cases.

## Podiatry treatment

Podiatry treatment was documented for 23 participants. Primary treatment included: nail and/ or skin care (n = 13); advice (n = 4); footwear/voucher provision (n = 3); referral for x-ray and GP to order orthopaedic shoe (n = 1); replacement of wound dressing (n = 1); and issuing of an orthotic (bunion night splint n = 1). Additional treatments included wound/blister care (n = 1), footwear (n = 2), advice (on diabetes management n = 1, self care n = 1 and elastic stockings n = 1), and issuing of off the shelf orthotics (n = 1).

After treatment, 18 participants stated their feet felt better, with one stating they had no change, one stating their foot was still painful, and one requiring ongoing care for improvement to occur. Eleven participants had ongoing issues that needed other healthcare provider interventions (dermatologist/GP/Physio, having surgery, voucher for shoes, fungal infection treatment).

Foot health resources were provided to 22 participants, shown in Table 4.

Additional podiatry visits were recommended for 13 (60%) participants treated by the podiatrist, with one participant attending the podiatrist twice.

## Results: Implementation factors

### Reach and adoption

Overall, 52 adults who were identified as potentially benefitting from podiatry care over a 19 month period participated in the program. Thirty-three (63%) of the 52 participants with identified foot problems accessed podiatry.

The participants included in the study were not reflective of the broader Australian population experiencing homelessness. The study participants were predominantly male, while reflective of the Melbourne population experiencing homelessness, the Australian population has more women experiencing homelessness [27]. Further, the study participants were older, with a median age 41.5 years, compared to the largest adult population experiencing homelessness being aged 25–34 years [27]. Australian population data shows more people living in supported accommodation, staying temporarily with other households or living in boarding houses or severely crowded dwellings and only 7% were rough sleepers [27]. In the current study, a third of participants were rough sleepers, 12% lived in supported accommodation, and smaller numbers were in temporary lodgings. This indicates those with higher needs were included, however reach and adoption of the broader population experiencing homelessness was limited.

### Feasibility, acceptability and appropriateness

The uptake of the program by people experiencing homelessness was slow, indicating the original methods were less effective than anticipated. The main mechanism to raise awareness was through discussion of the program at the regular HPP nurse meetings; selected because it was the most accessible avenue to share information to HPP nurses, fitting into the existing service structure. The Team discussed other avenues to access people experiencing homelessness with foot health issues. Youth refuges were identified as a potential site and were subsequently included in the recruitment strategy. HPP regularly engage youth from these services, and The Team used additional incentives that were known to be successful: holding a foot health

information session with free pizza, offering vouchers for footwear with a free foot health assessment on site at the refuge.

The collaborators generously donated costs and access to resources at no cost to HPP and people experiencing homelessness, increasing access to those who sought it. The nurses and podiatrists conveyed that the strategies were feasible, acceptable and appropriate.

### Fidelity and sustainability

The strategies developed by The Team were implemented as planned. As each collaborating organisation provides services to individuals experiencing homelessness as part of their core business, the Happy Feet program enabled the organisational processes necessary to support identification of foot issues by nurses, referral to podiatry as well as sharing the foot health resources to be developed. These processes were added to the organisational processes as business as usual and required limited additional work by staff to be provided, supporting sustainability.

### Discussion

The development of the Happy Feet program involved active engagement of the three stakeholders and reached 52 adults who might benefit from podiatry care over a 19 month period. Thirty-three (63%) of the 52 participants with identified foot problems accessed podiatry, indicating that the adoption of the whole program could be improved. People experiencing homelessness were less likely to access podiatry if they were born outside of Australia, lived in more precarious housing, had slightly more predisposing factors for requiring HPP nursing care and had fewer medical, psychological and cognitive conditions.

Implementing any new program into practice is not a simple feat. Research shows that it can take approximately 17 years for evidence-based interventions to get into practice [28]. Implementation science researchers have identified that this is due to the multiple factors necessary to enable new practices to occur [29]. There are many implementation frameworks and guides to support implementation, and this study selected frameworks with a focus on behaviour change, its clarity and practicality [19, 20], as well as engaging the key stakeholders across the whole journey to bests address the barriers and supported the enablers [21] in efforts to increase uptake by people experiencing homelessness. These activities were found to be feasible to undertake by the collaborating organisations and can be incorporated into the program ongoing. Upscaling the approach to other similarly structured services that support people experiencing homelessness is predicated on having key stakeholders that have a developed trusting relationship between both the collaborators and with people experiencing homelessness [30]. Relationships and trust are key in successful implementation [30]. Of note, there are few podiatry services in operation with expertise to build relationships and trust with people experiencing homelessness, limiting expansion of this type of collaboration to areas in which these services are in operation. Undertaking upscaling in larger and more complex organisations would require understanding the local context and identifying barriers and enablers and developing strategies to address and support them, respectively; context matters [31].

Part of implementation consists of reaching all of the target group, including those individuals who are traditionally seldom engaged. People experiencing homelessness are a group that experience significant barriers to accessing health services [22]; this study shows that there are sub-groups within this population who are even less likely to access health services. In the general Australian population, access to health care is reduced for different groups, including but not limited to people from culturally and linguistically diverse backgrounds, people with mental illness, people of low socioeconomic status and people living with a disability [32].

Individuals experiencing homelessness are generally considered as a homogenous group [33]. However many people experiencing homelessness often have more than one factor leading to structural disadvantage, with different combinations of disadvantage in different individuals [33]. There are many social determinants of health that influence health outcomes, such as the conditions in which people are born, grow, live work and age, with structural drivers of inequitable distribution of power, money and resources [34]. More research investigating the different characteristics of structural disadvantage of people experiencing homelessness, the intersectionality of these characteristics, and specific access needs for people experiencing intersectionality of structural disadvantage is required [33].

Despite the focus on supporting access to foot health resources in this study, the recruitment of people experiencing homelessness into the program was slow and over a third of participants (n = 19, 37%) did not access podiatry. People experiencing homelessness generally have limited resources and low agency to access the services they require to promote improved health and wellbeing [22, 35]. Barriers to access foot care services include the limited availability of affordable and accessible foot care services, embarrassment of individuals about their socioeconomic status or their foot condition, and the lack of permanent and affordable housing options which prevent people experiencing homelessness from seeking care [7, 9]. In the current project, nurses approached individuals with whom they had developed a relationship, reaching out to them to identify whether they had any foot health issues. The nurses then helped the participants to access foot health care if both the nurse and participant thought it would benefit participants. In previous research, although this group are difficult to engage, community health nurses have developed a significant trusting relationship that has led to people experiencing homelessness to engage with support services [23]. However, in our current study, not all participants who had been identified as having a foot health need by their nurse accessed the podiatrist for foot health care.

Part of the explanation may lie with the complexity of people experiencing homelessness. The HPP gathers data on the people seen as part of the service to better understand how to best support them, including predisposing factors for homelessness, health and social care information. Predisposing factors for homelessness is a domain that includes whether the individuals seen by the HPP nurses have issues in the areas of: intellectual, behavioural, alcohol and drug, social and mental health, factors that have been identified as important [18]. Participants in the current study had between one and six predisposing factors for homelessness, with 45 (87%) having two or more predisposing factors, and one participant having six predisposing factors for homelessness. In addition, multiple physical, mental and cognitive issues are also common in this group. Over half of the participants (56.9%) had at least two physical, mental or cognitive issues, with the one participant having 20 issues. This is a common finding in studies involving people experiencing homelessness [36, 37]. Therefore, while poor foot health appears to be an area of need in this group, and both the nurse and participant identified this need when together, compared to the many other physical, mental and cognitive issues participants are dealing with, foot health may not be the highest priority. As a result, the participant may pursue other issues once leaving the nurse. Further, despite the collaboration supporting easy access to foot health care, the complex psychosocial problems experienced by individuals experiencing homelessness may not have overcome the delays to accessing care until a crisis point is reached and their issue has become urgent [22]. It becomes incredibly difficult to think about managing a health problem when you don't know where you are going to sleep that night [22]. Consideration of how to best support access to foot care when an issue has been identified is needed. To help with this, a foot care model to determine the risk of foot problems has recently been developed for a Canadian setting with a view to identify those most at risk of foot problems early [8]. Further research is needed to further develop this

model and investigate how to support participants at risk of foot problems to access podiatry before their foot health issue reaches crisis point.

Another foot care access issue involves ongoing treatment for foot problems. In the current study, of the 23 participants treated by the podiatrist, additional visits were recommended for 57% of them. Research on access to care by people experiencing homelessness in general practice identified that single episodes of care is common, and often the podiatrist is only seen because a podiatrist is on site when the individual has attended a drop-in medical clinic [38]. Another area of future work is to consider how to support individuals experiencing homelessness to access multiple episodes of care so that their foot problem is managed appropriately. Untreated foot complications may lead to the development of more serious foot problems, particularly in those individuals experiencing homelessness who have diabetes, and could potentially necessitate hospitalisation. Further research on the cost-effectiveness of providing foot care interventions in people experiencing homelessness is warranted. When comparing access to foot care for other population groups, such as people with diabetes who are at particularly high risk of serious foot complications, there has been limited research to incorporate implementation science approaches to improve access to screening or management of foot complications. Currently, only around 50% of people with diabetes undergo a foot screening to ascertain risk of serious foot complications in the general population [39, 40]. Research to identify barriers and enablers to delivering preventative and early intervention footcare has been undertaken [41], and future work can use these findings to develop programs to increase uptake of foot screening followed by evidence-based management in this population group.

## Strengths and limitations

A strength of this study is that it included prospective data collection involving individuals experiencing homelessness. Comprehensive data describing the whole person was included, with a specific focus on foot health to describe the population.

It is important to note that recruitment for this study was going to be for 12 months from May 2019. It was incredibly difficult to recruit a greater number of participants for this study. By May 2020, only 44 participants were recruited. The study team sought to recruit at least 50 participants, considering this number a minimum number to provide useful information about the population, their foot health and service usage, therefore extended recruitment until the end of 2020 to attempt to meet this target. Unfortunately, COVID-19 significantly impacted on the community in Melbourne from April 2020, with high levels of restrictions on movements and contacts, and increased burden on staff so that only eight more participants were recruited during the additional period, mainly through the addition of accessing a youth refuge. We did not have the opportunity to collect data on the number of referrals to the podiatrist from the HPP nurses prior to the study, limiting the ability to ascertain whether the impact of the interventions made a difference in referrals. It is likely that more individuals than those who participated in this study had foot problems and would have benefited from podiatry. Therefore, this population group may not accurately represent the actual population of people experiencing homelessness in Melbourne. Another potential limitation is that the results may not be generalisable to other contexts that differ in respect to healthcare access, community support availability and the nature of local governance. As discussed earlier, context matters [31], and organisations of different sizes or complexity would need to identify their local barriers and enablers and develop strategies to address them to enable successful implementation. While including a number of key stakeholders, the study did not collect any information on the barriers to accessing foot care by people experiencing homelessness. This study would have benefited from also including people experiencing homelessness in the

development and implementation of the program. This may have improved uptake of the program by people experiencing homelessness. Any engagement would need to include a range of individuals that reflect the heterogeneity of this population group to ensure that the input obtained can guide development of services that truly meets the needs of the target population.

Future research could focus on understanding what is needed to further support access to health care with a view to prevent the development serious foot heath issues in people experiencing homelessness and incorporate their insights in program development and implementation.

## Conclusion

A multi-pronged, structured program involving three collaborating organisations was implemented where nurses identified then referred people experiencing homelessness with foot health needs to podiatry and/or provided foot health resources. The implementation methods and intervention were delivered as planned. Even with this support to access foot health resources, 37% of participants did not access podiatry. Future work is needed to understand what is needed to further support access to health care of people experiencing homelessness, with a view to prevent the development serious foot heath issues. Studies on cost effectiveness of programs to address foot health in people experiencing homelessness are also warranted.

## Supporting information

**S1 File. StaRI checklist.**
(DOCX)

**S2 File. Primary ethics committee approval document.**
(PDF)

**S3 File. Secondary ethics committee approval email.**
(PDF)

**S4 File. Human participants research checklist.**
(DOCX)

## Acknowledgments

All the Homeless Persons Program client participants and nurses for contributing to making this project possible.

Youth refuges, Vicki's Place and Hope Street, and all the people who are at risk or experiencing homelessness who participated, and all the staff who helped make this project possible.

## Author Contributions

**Conceptualization:** Rajna Ogrin, Mary-Anne Rushford, Anthony Lewis.

**Data curation:** Rajna Ogrin, Joseph Fallon, Rebecca Mannix, Anthony Lewis.

**Formal analysis:** Rajna Ogrin.

**Investigation:** Rajna Ogrin.

**Methodology:** Rajna Ogrin.

**Project administration:** Rajna Ogrin, Mary-Anne Rushford, Joseph Fallon, Ben Quinn, Anthony Lewis.

**Resources:** Rajna Ogrin, Mary-Anne Rushford, Rebecca Mannix, Ben Quinn, Anthony Lewis.

**Writing – original draft:** Rajna Ogrin.

**Writing – review & editing:** Rajna Ogrin, Mary-Anne Rushford, Joseph Fallon, Rebecca Mannix, Ben Quinn, Anthony Lewis.

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
