## [Decision Letter · Decision Letter 0]

14 Feb 2024

PONE-D-24-01451Describing the development and implementation of a novel collaborative multidisciplinary approach to deliver foot health supports for individuals experiencing homelessness and its outcomes.PLOS ONE

Dear Dr. Ogrin,

Thank you for submitting your manuscript to PLOS ONE. After careful consideration, we feel that it has merit but does not fully meet PLOS ONE’s publication criteria as it currently stands. Therefore, we invite you to submit a revised version of the manuscript that addresses the points raised during the review process.

Thank you for submitting this excellent manuscript. Please address a few minor comments from the reviewers.

We look forward to receiving your revised manuscript.

Kind regards,

Tze-Woei Tan, M.D.

Academic Editor

PLOS ONE

Journal Requirements:

2. In this instance it seems there may be acceptable restrictions in place that prevent the public sharing of your minimal data. However, in line with our goal of ensuring long-term data availability to all interested researchers, PLOS’ Data Policy states that authors cannot be the sole named individuals responsible for ensuring data access (http://journals.plos.org/plosone/s/data-availability#loc-acceptable-data-sharing-methods).

Reviewers' comments:

Reviewer's Responses to Questions

**Comments to the Author**

1. Is the manuscript technically sound, and do the data support the conclusions?

Reviewer #1: Yes

Reviewer #2: Yes

2. Has the statistical analysis been performed appropriately and rigorously? 

Reviewer #1: Yes

Reviewer #2: Yes

3. Have the authors made all data underlying the findings in their manuscript fully available?

Reviewer #1: Yes

Reviewer #2: No

4. Is the manuscript presented in an intelligible fashion and written in standard English?

Reviewer #1: Yes

Reviewer #2: Yes

5. Review Comments to the Author

Reviewer #1: Kudos to the authors for a very commendable program within an underserved population. This manuscript makes for an unconventional and interesting read.

Will be good for the authors to comment on the funding sources to the program (if any) and long-term sustainability.

The study participants will likely require as much, if not more social services support than foot health services. Would the authors be able to discuss on the socio-economical determinants of health?

Reviewer #2: Excellent project. I congratulate the authors on this new data for the clinical and research community.

I am trained in implementation science, and a researcher in limb preservation. My comments and suggestions are intended to be constructive.

Title: why multidisciplinary and not interdisciplinary?

Homeless person need to be empowered, engage in such a program.

Why not use co-design, to ensure that the program corresponds as closely as possible to the needs of homeless people?

Discussion :

It would be interesting to hear about the possibility of scaling this type of program in the future, given that this seems to be a pilot implementation.

6. PLOS authors have the option to publish the peer review history of their article (what does this mean?). If published, this will include your full peer review and any attached files.

Reviewer #1: No

Reviewer #2: No

---

## [Author Response · Author response to Decision Letter 0]

6 Mar 2024

Please see authors responses to the editor and reviewers in the response to reviewers attachment. Thank you.

---

## [Decision Letter · Decision Letter 1]

9 Apr 2024

Describing the development and implementation of a novel collaborative multidisciplinary approach to deliver foot health supports for individuals experiencing homelessness and its outcomes.

PONE-D-24-01451R1

Dear Dr. Ogrin,

We’re pleased to inform you that your manuscript has been judged scientifically suitable for publication and will be formally accepted for publication once it meets all outstanding technical requirements.

Kind regards,

Tze-Woei Tan, M.D.

Academic Editor

PLOS ONE

Additional Editor Comments (optional):

Reviewers' comments:

Reviewer's Responses to Questions

**Comments to the Author**

1. If the authors have adequately addressed your comments raised in a previous round of review and you feel that this manuscript is now acceptable for publication, you may indicate that here to bypass the “Comments to the Author” section, enter your conflict of interest statement in the “Confidential to Editor” section, and submit your "Accept" recommendation.

Reviewer #1: All comments have been addressed

2. Is the manuscript technically sound, and do the data support the conclusions?

Reviewer #1: Yes

3. Has the statistical analysis been performed appropriately and rigorously? 

Reviewer #1: Yes

4. Have the authors made all data underlying the findings in their manuscript fully available?

Reviewer #1: Yes

5. Is the manuscript presented in an intelligible fashion and written in standard English?

Reviewer #1: (No Response)

6. Review Comments to the Author

Reviewer #1: Thank you for your revision. Keep you the good work and looking forward to future results from your research

7. PLOS authors have the option to publish the peer review history of their article (what does this mean?). If published, this will include your full peer review and any attached files.

Reviewer #1: No
